# Trust-Aware Planning:
# Modeling Trust Evolution in Longitudinal Human-Robot Interaction

**Zahra Zahedi, Mudit Verma, Sarath Sreedharan, Subbarao Kambhampati**

SCAI, Arizona State University
{zzahedi, mverma13, ssreedh3, rao}@asu.edu

## Abstract

Trust between team members is an essential requirement for any successful cooperation. Thus, engendering and maintaining the fellow team members' trust becomes a central responsibility for any member trying to not only successfully participate in the task but to ensure the team achieves its goals. The problem of trust management is particularly challenging in mixed human-robot teams where the human and the robot may have different models about the task at hand and thus may have different expectations regarding the current course of action and forcing the robot to focus on the costly explicable behavior. We propose a computational model for capturing and modulating trust in such longitudinal human-robot interaction, where the human adopts a supervisory role. In our model, the robot integrates human's trust and their expectations from the robot into its planning process to build and maintain trust over the interaction horizon. By establishing the required level of trust, the robot can focus on maximizing the team goal by eschewing explicit explanatory or explicable behavior without worrying about the human supervisor monitoring and intervening to stop behaviors they may not necessarily understand. We model this reasoning about trust levels as a meta reasoning process over individual planning tasks. We additionally validate our model through a human subject experiment.

## Introduction

Building and maintaining trust between team members form an essential part of any human teaming endeavor. We expect this characteristic to carry over to human-robot teams and the ability of an autonomous agent to successfully form teams with humans directly depends on their ability to model and work with human's trust. Unlike homogenous human teams, where the members generally have a well-developed sense of their team member's capabilities and roles, teaming between humans and autonomous agents may suffer because of the user's misunderstanding about the robot's capabilities. Thus the understanding and (as required) correction of the human's expectations about the robot can be a core requirement for engendering lasting trust from the human teammate. Recent works in human-aware planning, particularly those related to explicable planning (Zhang et al. 2017) and generating model reconciliation (Chakraborti et al. 2017), can provide us with valuable tools that can empower autonomous agents to shape the user's expectation correctly and by extension, their trust.

In this paper, we will consider one of the most basic human-robot teaming scenarios, one where the autonomous agent is performing the task and the human is following a supervisory role. For this setting, we propose a meta-computational framework that can model and work with the user's trust in the robot to correctly perform its task. We will show how this framework allows the agent to reason about the fundamental trade-off between (1) the more expensive but trust engendering behavior, including explicable plans and providing explanations, and (2) the more efficient but possibly surprising behavior the robot is capable of performing. Thus our framework is able to allow the agent to take a long term view of the teaming scenario, wherein at earlier points of teaming or at points with lower trust, the agent is able to focus on trust-building behavior so that later on, it can use this engendered trust to follow more optimal behavior. We will validate this framework by demonstrating the utility of this framework on a modified rover domain and also perform a user study to evaluate the ability of our framework to engendering trust and result in higher team utility.

## Related Work

There exists a number of works that have studied trust in the context of human-robot interaction. The works in this area can be broadly categorized into two groups (1) Trust inference based on observing human behavior or (2) Utilizing estimated trust to guide robot behavior.

For Trust inference, Online Probabilistic Trust Inference Model (OPTIMo) is one of the pioneers in this area in which they capture trust as a latent variable represented with a dynamic Bayesian network. OPTIMo uses a technique for estimating trust in real-time that depends on the robot's task performance, human intervention, and trust feedback (Xu and Dudek 2015). Trust inference model based on Bayesian inference with Beta-distribution to capture both positive and negative attitude on robot's performance (Guo, Zhang, and Yang 2020) contributes an important extension to OPTIMo. Also, this Bayesian reasoning for trust inference has been considered non-parametrically with Gaussian processes, Recurrent Neural Network (RNN), and a hybrid approach in which trust is a task-dependent latent function (Soh et al. 2020).

With regards to trust utilization, some works try to esti-

mate trust, given human intervention and robot command, using reputation function (Xu and Dudek 2012), or OP-TIMO (Xu and Dudek 2016) to make an adaptive mechanism that dynamically adjusts the robot's behaviors, to improve the efficiency of the collaborative team. Also, an extension of OPTIMo with time series trust model (Wang et al. 2015) has been used to estimate trust in multi-robot scenarios. The estimated trust is utilized to decide between manual or autonomous control mode of robots (Wang et al. 2018). In (Chen et al. 2018, 2020), a POMDP planning model has been proposed that allows the robot to obtain a policy by reasoning about human's trust as a latent variable. In swarm robots, they leveraged trust to update the communication graph that will reduce the misleading information from less trusted swarm robots (Liu et al. 2019).

This paper is situated in the trust utilization area since the robot is trying to use trust to make a meta planning decision. Although most of the mentioned work tried to utilize trust for better team performance, they all used trust in the action level and didn't consider how the trust will affect robot performance at the problem level. As they consider trust as a tool to improve performance in cooperation, the importance of considering trust that comes with more interpretable behavior has been neglected in those works.

## Background

In this section we will introduce some of the basic concepts related to planning that we will be using to describe our framework.

**A Classical Planning** problem is $\mathcal{M} = \langle \mathcal{D}, \mathcal{I}, \mathcal{G} \rangle$ where $\mathcal{D} = \langle F, A \rangle$ is a domain with $F$ as a set of fluents that define a state $s \subseteq F$, also initial $\mathcal{I}$ and goal $\mathcal{G}$ states are subset of fluent $\mathcal{I}, \mathcal{G} \subseteq F$, and each action in $a \in A$ is defined as follows $a = \langle c_a, pre(a), eff^{\pm}(a) \rangle \in A$, where $A$ is a set of actions, $c_a$ is the cost, and $pre(a)$ and $eff^{\pm}$ are precondition and add or delete effects. i.e. $\rho_{\mathcal{M}}(s, a) \models \perp$ $if \ s \not\models pre(a); \ else \ \rho_{\mathcal{M}}(s, a) \models s \cup eff^+(a) \setminus eff^-(a)$, and $\rho_{\mathcal{M}}(.)$ is the transition function.
So, when we talk about model $\mathcal{M}$, it consists of action model as well as initial state and goal state. The solution to the model $\mathcal{M}$ is a plan which is a sequence of actions $\pi = \{a_1, a_2, \dots, a_n\}$ which satisfies $\rho_{\mathcal{M}}(\mathcal{I}, \pi) \models \mathcal{G}$. Also, $C(\pi, \mathcal{M})$ is the cost of plan $\pi$ where

$$C(\pi, \mathcal{M}) = \begin{cases} \sum_{a \in \pi} c_a & if \ \rho_{\mathcal{M}}(\mathcal{I}, \pi) \models \mathcal{G} \\ \infty & o.w \end{cases}.$$

**Human-Aware planning** (HAP) in its simplest form consists of scenarios, where a robot is performing a task problem and human is observing and evaluating the task. So it can be defined by a tuple of the form $\langle \mathcal{M}^R, \mathcal{M}_h^R \rangle$, where $\mathcal{M}_R$ is the planning problem being used by the robot and $\mathcal{M}_h^R$ is the human's understanding of the task (which may differ from the robot's original model). They are defined as $\mathcal{M}^R = \langle \mathcal{D}^R, \mathcal{I}^R, \mathcal{G}^R \rangle$ and $\mathcal{M}_h^R = \langle \mathcal{D}_h^R, \mathcal{I}_h^R, \mathcal{G}_h^R \rangle$.
So, in general, the robot is expected to solve the task while meeting the user's expectations. As such, for any given plan, the degree to which the plan meets the user expectation is measured by the explicability score of the plan, which is de-

fined to be the distance ($\delta$) between the current plan and the plan expected by the user ($\pi^E$).

$$E(\pi) = -1 * \delta(\pi^E, \pi)$$

We will refer to the plan as being perfectly explicable when the distance is zero. A common choice for the distance is the cost difference in the human's model for the expected plan and the optimal plan in the human model (Kulkarni et al. 2019). Here the robot has two options, (1) it can choose from the possible plans it can execute the one with the highest explicability score (referred to as the explicable plan), or (2) it could try to explain, wherein it updates the human model through communication, to a model wherein the plan is chosen by the robot is either optimal or close to optimal and thus have a higher explicability score (Sreedharan et al. 2020a; Chakraborti, Sreedharan, and Kambhampati 2017). A form of explanation that is of particular interest, is what's usually referred to as a *minimally complete explanation* or MCE (Chakraborti et al. 2017), which is the minimum amount of model information that needs to be communicated to the human to make sure that the human thinks the current plan is optimal. In the rest of the paper, when we refer to explanation or explanatory messages, we will be referring to a set of model information (usually denoted by $\varepsilon$), where each element of this set corresponds to some information about a specific part of the model. We will use $+$ operator to capture the updated model that the human would possess after receiving the explanation. That is, the updated human model after receiving an explanation $\varepsilon$ will be given by $\mathcal{M}_h^R + \varepsilon$.

**A Markov Decision Process** (MDP) is $\langle S, A, C, P, \gamma \rangle$ where $S$ denote the finite set of states, $A$ denotes the finite set of actions, $C : S \times A \to \mathbb{R}$ is a cost function, $P : S \times S \times A \to [0 \ 1]$ is the state transition function and $\gamma$ is the discount factor where $\gamma \in [0 \ 1]$. An action $a$ at state $s_n$ at time $n$ incurs a cost $(s_n, a)$ and a transition $P(s_n, s_{n+1}, a)$ where $s_{n+1}$ is the resulting state which satisfies Markov property. So, the next state only depends on the current state and the action chosen at the current state. A policy $\pi(s)$ denotes as action chosen at state $s$. The problem in an MDP is to find an optimal policy $\pi : S \to A$ that maximizes the cumulative cost function (please note that the cost function here is defined as a negative of costs). Over a potentially infinite time horizon, we need to maximize the expected discounted costs $\sum_{n=0}^{\infty} \gamma^k C(S, R)$.

## Problem Definition

We will focus on a human-robot dyad, where the human (H) adopts a supervisory role and the robot is assigned to perform tasks. We will assume that the human's current level of trust is an approximate discretization of a continuous value between 0 to 1, and it can be mapped to one of the sets of ordered discrete trust levels. We will assume that the exact problem to be solved at any step by the robot is defined as a function of the current trust the human has in the robot, thereby allowing us to capture scenarios where the human may choose to set up a trust-based curriculum for the robot to follow. In particular, we will assume that each trust level

is associated with a specific problem, which is known to the robot *a priori*, thereby allowing for precomputation of possible solutions. In general, we expect the human's actions to be completely determined by their trust in the robot, and we will model the robot's decision-making level as two levels decision-making process. Before describing the formulation in more detail, let us take a quick look at some of the assumptions we are making regarding the problem setting and clarify our operational definition of trust.

## Assumptions

**Robot (R)**, is responsible for executing the task.

1. Each task is captured in the robot model by a deterministic, goal-directed model $\mathcal{M}^R$ (which is assumed to be correct). The robot is also aware of the human's expected model of the task $\mathcal{M}_h^R$ (which could include the human's expectation about the robot). As assumed in most HAP settings, these models could differ over any of the dimensions (including action definitions, goals, current state, etc.).

2. For simplicity, we will assume that each task assigned is independent of each other, in so far that no information from earlier tasks is carried over to solve the later ones.

3. The robot has a way of accessing or identifying the current state of the human supervisor's trust in the robot. Such trust levels may be directly provided by the supervisor or could be assessed by the robot by asking the human supervisor specific questions.

**Human (H)**, is the robot's supervisor and responsible for making sure the robot will perform the assigned tasks and will achieve the goal.

1. For each problem, the human supervisor can either choose to monitor ($ob$) or not monitor ($\neg ob$) the robot.

2. Upon monitoring the execution of the plan by R, if H sees an unexpected plan, they can intervene and stop R.

3. The human's monitoring strategy and intervention will be completely determined by the trust level. With respect to the monitoring strategy, we will assume it can be captured as a stochastic policy, such that for a trust level $i$ the human would monitor with a probability of $\omega(i)$. Moreover, the probability of monitoring is inversely proportional to the level of trust. In terms of intervention, we will assume that the lower the trust and the more unexpected the plan, the earlier the human would end the plan execution. We will assume the robot has access to a mapping from the current trust level and plan to when the human would stop the plan execution.

## Human Trust and Monitoring strategy

Before going further, let us examine the exact definition of the trust we will rely on. According to a widely accepted trust definition, *trust is a psychological state comprising the intention to accept vulnerability based upon the positive expectations of the intentions or behavior of another* (Rousseau et al. 1998). So, according to this definition, when we have human-robot interaction, the human can choose to be vulnerable by 1) Not intervening in the robot's actions while it is doing something unexpected and 2) Not to monitor the robot while the robot might do inexplicable behavior (Sengupta, Zahedi, and Kambhampati 2019). Thus, a human with a high level of trust in the robot would expect the robot to achieve their goal and as such, might choose not to monitor the robot, or even if they monitor and the robot may be performing something unexpected, they are less likely to stop the robot (they may trust the robot's judgment and may believe the robot may have a more accurate model of the task). Thus, when the trust increases, it is expected that the human's monitoring and intervention rate decreases. We can say monitoring rate, as well as intervention rate being a function of the current trust (even being inversely proportional). So, given the trust level human has on the robot, the robot can reason about the monitoring and intervention rate of the human supervisor.

## Base Decision-Making Problem

As mentioned earlier, here, each individual task assigned to the robot can be modeled as a human-aware planning problem of the form $\langle \mathcal{M}^R, \mathcal{M}_h^R \rangle$. Now given such a human-aware planning problem, the robot has the following options.

1. In the simplest case, the robot could choose to execute either its explicable plan ($\pi_{exp}$) or its optimal plan ($\pi_{opt}$). Such that the cost of executing the explicable plan is guaranteed to be greater than or equal to the cost of the optimal plan , $C_e(\pi_{exp}) \geq C_e(\pi_{opt})$, where $C_e(\pi) = C(\pi, \mathcal{M}^R)$ is the cost of executing the plan in $\mathcal{M}^R$.

2. Now, if the robot chooses to follow its optimal plan, then it could augment that plan with an explanation (which is expected to be provided upfront before the plan gets executed). Now the robot could choose to provide either an MCE $\varepsilon^{MCE}$, or an explanation that merely increases the explicability of a trace and doesn't guarantee that the plan would be optimal in the updated human model. We will denote such explanations as $\tilde{\varepsilon}$. The cost of following such a strategy for a robot is given as $C_e(\langle \varepsilon, \pi \rangle) = C(\varepsilon) + C(\pi, \mathcal{M}^R)$, where $C(\varepsilon)$ is the cost of communicating the explanation.

To simplify the discussion, we will assume that for each trust level, the robot has to perform a fixed task. So if there are $k$-levels of trust, then the robot would be expected to solve $k$ different tasks. Moreover, if the robot is aware of these tasks in advance, then it would be possible for it to precompute solutions for all these tasks in advance and make the choice of following one of the specific strategies mentioned above depending on the human's trust and the specifics costs of following each strategy.

## Meta-MDP Problem

Next, we will talk about the decision-making model we will use to capture the longitudinal reasoning process the robot will be following to decide what strategy to use for each task. The decision epochs for this problem correspond to the robot getting assigned a new problem. The cost structure of this meta-level problem includes not only the cost incurred by the robot in carrying out the task but team level costs

related to the potential failure of the robot to achieve the goal, how the human supervisor is following a specific monitoring strategy, etc. Specifically, we will model this problem as an infinite horizon discounted MDP of the form $\mathbb{M} = \langle \mathbb{S}, \mathbb{A}, \mathbb{P}, \mathbb{C}, \gamma \rangle$, defined over a state space consisting of $k$ states, where each state corresponds to the specific trust level of the robot. Given the assumption that each of the planning tasks is independent, the reasoning at the meta-level can be separated from the object-level planning problem. In this section, we will define this framework in detail, and in the next section, we will see how such framework could give rise to behavior designed to engender trust.

**Meta-Actions** $\mathbb{A}$**:** Here the robot has access to four different actions, corresponding to four different strategies they can follow, namely, use the optimal plan $\pi_{opt}$, the explicable plan $\pi_{exp}$, follow $\pi_{opt}$ while providing an explanation that improves the explicability score $\langle \tilde{\varepsilon}, \pi_{opt} \rangle$ and finally providing MCE for the optimal plan $\langle \varepsilon^{MCE}, \pi_{opt} \rangle$.

**Transition Function** $\mathbb{P}$**:** The transition function captures the evolution of the human's trust level based on the robot's action. In addition to the choice made by the robot, the transition of the human trust also depends on the user's monitoring strategies, which we take to be stochastic but completely dependent on the human's current level of trust and thus allowing us to define a markovian transition function. In this model, for any state, the system exhibits two broad behavioral patterns, the ones for which the plan is perfectly explicable in the (potentially updated) human model and for those in which the plan may not be perfectly explicable.

- Perfectly explicable plan: The first case corresponds to one where the robot chooses to follow a strategy the human accepts to be optimal. Here we expect the human trust to increase to the next level in all but the maximum trust level (where it is expected to remain the same). The most common case where this may happen is when the robot chooses to provide an MCE explanation. Though there may also be cases where the explicable plan also perfectly matches up with the human's expected plan.
- Other Cases: In this case, the robot chooses to follow a plan with a non-perfect explicability score $E(\pi)$. Now for any level that is not the maximum trust level, this action could cause a transition to one of three levels, the next trust level $s_{i+1}$, stay at the current level $s_i$, or the human could lose trust on the robot and move to level $s_{i-1}$. Here the probabilities for these three cases for a meta-level action associated with a plan $\pi$ are as given below

$$P(s_i, a^\pi, s_{i+1}) = (1 - \omega(i))$$

where $\omega(i)$ is the probability that the human would choose to observe the robot at a trust level $i$. Thus for a non-explicable plan, the human could still build more trust in the robot if they notice the robot had completed its goal and had never bothered monitoring it.

$$P(s_i, a, s_i) = \omega(i) * \mathcal{P}(E(\pi))$$

That is, the human's trust in the robot may stay at the same level even if the human chooses to observe the robot. Note the probability of transition here is also dependent on a function of the explicability score of the current plan, which is expected to form a well-formed probability distribution ($\mathcal{P}(\cdot)$). Here we assume this is a monotonic function over the plan explicability score; a common function one could adopt here is a Boltzmann distribution over the score (Sreedharan et al. 2020b). For the maximum trust level, we would expect the probability of staying at the same level to be the sum of these two terms. With the remaining probability, the human would move to a lower level of trust.

$$P(s_i, a, s_{i-1}) = \omega(i) * (1 - \mathcal{P}(E(\pi)))$$

**Cost function** $\mathbb{C}$**:** For any action performed in the meta-model, the cost function ($\mathbb{C} : \mathbb{S} \times \mathbb{A} \rightarrow \mathcal{R}$) depends on whether the human is observing the robot or not. Since we are not explicitly maintaining state variables capturing whether the human is monitoring, we will capture the cost for a given state action pair as an expected cost over this choice. Note that the use of this simplified cost model does not change the optimal policy structure as we are simply moving the expected value calculation over the possible outcome states into the cost function. Thus the cost function becomes

$$\mathbb{C}(s_i, a^\pi) = (1 - \omega(i)) * (C_e(\pi)) + \omega(i) * C_{\langle \mathcal{M}^R, \mathcal{M}_h^R \rangle}$$

Where $C_e(\pi)$ is the full execution cost of the plan (which could include explanation costs) and the $C_{\langle \mathcal{M}^R, \mathcal{M}_h^R \rangle}$ represents the cost of executing the selected strategy under monitoring. For any less than perfectly explicable plan, we expect the human observer to stop the execution at some point, and as such, we expect $C_{\langle \mathcal{M}^R, \mathcal{M}_h^R \rangle}$ to further consist of two cost components. The cost of executing the plan prefix till the point of intervention by the user and the additional penalty of not completing the goal.

**Discounting** $\gamma$**:** Since in this setting, higher trust levels are generally associated with higher expected values, one could adjust discounting as a way to control how aggressively the robot would drive the team to higher levels of trust. With lower values of discounting favoring more rapid gains in trust.

**Remark:** One central assumption we have made throughout this paper is that the robot is operating using the correct model of the task (in so far as it is correctly representing the true and possibly unknown task model $\mathcal{M}^*$). As such, it is completely acceptable to work towards engendering complete trust in the supervisor, and the human not monitoring the robot shouldn't lead to any catastrophic outcome. Obviously, this need not always be true. In some cases, the robot may have explicit uncertainty over how correct its model is (for example, if it learned this model via Bayesian methods), or the designer could explicitly introduce some uncertainty into the robot's beliefs about the task (this is in some ways parallel to the recommendations made by the

off-switch game paper (Hadfield-Menell et al. 2016) in the context of safety). In such cases, the robot would need to consider the possibility that when the human isn't observing, there is a small probability that it will fail to achieve its task. One could attach a high negative reward to such scenarios, in addition to a rapid loss of trust from the human. Depending on the exact probabilities and the penalty, this could ensure that the robot doesn't engender complete trust when such trust may not be warranted (thereby avoiding problems like automation bias (Cummings 2004)).

## Evaluation and Implementation

This section will describe a demonstration of our framework in a modified rover domain instance and describe a user study we performed to validate our framework. Throughout this section, we will use the following instantiation for the framework. We considered 4 states. For each of these trust states, we associate a numerical value $T(i) \in [0 \ 1]$, that we will use to define the rest of the model. Specifically, the $T(i)$ values we used per state were 0, 0.3, 0.6 and 1 respectively. For monitoring strategy, we used $\omega(i)$ as a Bernoulli distribution with probability of $(1 - T(i))$, as explicability score $E(\pi)$ we used the negative of the cost difference between the current plan and the optimal plan in the robot model. For $\mathcal{P}(\cdot)$, we have 1 for the explicable plan and 0 for the optimal plan. For execution cost, we assumed all actions are unit cost. We will ignore explanations in the experiments and focus on cases where the choice of the robot is limited between explicable and optimal plans.

**Implementation** We implemented our framework using Python which was run on an Ubuntu workstation with an Intel Xeon CPU (clock speed 3.4 GHz) and 128GB RAM. We used Fast Downward with A* search and the lmcut heuristic (Helmert 2006) to solve the planning problems and find the plans in all 4 problems, then we used the python MDP-toolbox (Cordwell 2012) to solve the meta-MDP problem for the robot's meta decision. The total time for solving the base problem was $0.0125s$ when applicable and $0.194s$ for solving the meta-MDP problem.

### Rover Domain Demonstration

Here, we used the updated version of IPC[1] Mars Rover (The Rover (Meets a Martian) Domain) in (Chakraborti, Sreedharan, and Kambhampati 2017) and changed it a little by adding metal sampling to the domain as well. In the Rover (Meets a Martian) Domain, it is assumed that the robot can carry soil, rock, and metal at the same time and doesn't need to empty the store before collecting new samples and the Martian (the supervisor in this scenario) isn't aware of this new feature. Also, the Martian believes that for the rover to perform take_image action; it needs to also send the soil and metal data collected from the waypoint from where it is taking the image. So the Martian's model of the rover has an additional precondition (empty ?s) for actions sample_soil,

---

[1] From the International Planning Competition (IPC) 2011: http://www.plg.inf.uc3m.es/ipc2011-learning/Domains.html

sample_rock, and sample_metal, and extra preconditions for the action take_image.

Now for each problem, the rover is expected to communicate soil, rock, metal, and images from a set of waypoints. Given the additional preconditions in the Martian model, the expected plan in the Martian model would be longer than what is required for the rover. For example, in the first problem, the rover goal consists of communicate_metal_data waypoint0 and communicate_metal_data waypoint3. For this problem, the explicable and optimal plan would be as follows
$\pi_{exp}^1 =$

```
(sample_metal rover0 rover0store waypoint3)
(communicate_metal_data rover0 general waypoint3 waypoint3 waypoint0)
(navigate rover0 waypoint3 waypoint0)
(drop rover0 rover0store)
(sample_metal rover0 rover0store waypoint0)
(navigate rover0 waypoint0 waypoint3)
(communicate_metal_data rover0 general waypoint0 waypoint3 waypoint0)
```

$\pi_{opt}^1 =$

```
(sample_metal rover0 rover0store waypoint3)
(communicate_metal_data rover0 general waypoint3 waypoint3 waypoint0)
(navigate rover0 waypoint3 waypoint0)
(sample_metal rover0 rover0store waypoint0)
(navigate rover0 waypoint0 waypoint3)
(communicate_metal_data rover0 general waypoint0 waypoint3 waypoint0)
```

For a set of four sample tasks from this domain, the meta-policy calculated by our system is as follows $\{\pi_{exp}^1, \pi_{exp}^2, \pi_{exp}^3, \pi_{opt}^4\}$. Note how the policy prescribes the use of the explicable plan for all but the highest level of trust, this is expected given the fact that the optimal plans here are inexecutable in the human model, and if the supervisor observes the robot following such a plan, it is guaranteed to lead to a loss of trust. The rover chooses to follow the optimal plan at the highest level since the supervisor's monitoring strategy at these levels is never to observe the rover. The expected value of this policy for the lowest level of trust is $-179.34$, while if the robot were to always execute the explicable plan, the value would be $-415.89$. Thus, we see that our trust-adaptive policy does lead to an improvement in the rover's total cost.

### Human Subject Experiment

To evaluate the performance of our system, we compared our method (**Trust-Aware** condition) against two baseline cases,
(1) **Always Explicable**: Under this condition, the robot always executes the plan, which is explicable to humans.
(2) **Random Policy**: Under this condition, the robot randomly executes the explicable or inexplicable plan.
In particular, we aim to evaluate the following hypotheses

**H1-** The team performance, i.e., the total cost of plan execution and human's monitoring cost in the trust-aware condition, will be better than the team performance in the always explicable condition.

**H2-** The level of trust engendered by the trust-aware condition will be higher than that achieved by the random policy.

**Experiment Setup** We designed a user interface that gamifies the human's decisions to monitor the robot or not. The

participants thus play the role of the supervisor and are responsible for making sure the robot is performing its assigned tasks and is achieving its goals. Each participant has 10 rounds of the robot doing tasks. Depending on the choices made by the participants, they either gain or lose points. They are told that they will be awarded 100 points if the robot does the task right and achieves the assigned goal. At the beginning of each round, they can either choose to monitor the robot to make sure it does its job[2] or they can choose to perform another task (thereby forgoing monitoring of the robot) to make extra points. In this case, the extra task was labeling images for which they will receive 100 points (in addition to the points they receive from the robot doing its tasks successfully). However, if they choose to label images, and the robot fails to achieve its goal, they *lose* 200 points ($-200$ points). Also, if they choose to monitor the robot, and they see the robot is doing something invalid or wrong, they can choose to stop the robot. If this happens, they only receive 50 points. But if they let the robot finish a potentially invalid plan, and if the robot couldn't achieve the goal at the end, then they again lose points ($-200$ points).

In this study, we again considered a curriculum of 4 trust levels and 4 different tasks for the robot. Each task consists of the robot operating on a grid map with different goals such as moving to a certain location and bringing coffee from a place to another place.[3] For each problem; the map is shown to the participants are different from the robot's map. As a result, the plan the human expects is different from the robot's optimal plan. Thus, in each task, the robot can either execute a costly but explicable plan or an optimal but inexplicable plan.

**Robot Tasks** Figure 1(a) presents the robot and human map corresponding to each of the tasks used in the study. For each map, we also highlight the explicable and optimal plan ($\pi_1$ and $\pi_2$ respectively) in the figure. As part of the study, each participant is presented with the human map and is told to expect the robot to follow the shortest possible path. Additionally, we also provide a key describing the semantics of each of the visual elements that are part of the map (as shown in Figure 1(b)). The tasks are as follows.

**Task 1.** The robot task is to reach the red point on the map

**Task 2.** The robot task is to bring coffee to the room

**Task 3.** The robot task is to bring coffee to the room[4]

**Task 4.** The robot task is to take coffee from room 1 and bring it to room 2

For each round, if the participant chooses to monitor, a step-by-step plan execution is shown to them and with an option to stop the robot at any step. At the end of each round, a four-item trust scale of Muir questionnaire (Muir 1994) is given to them, which measures their trust in that

---

[2]Their primary responsibility is to ensure the robot completes its task

[3]All the details of user experiment setup, including the tasks are provided in the supplementary materials

[4]Tasks 2 and 3 are different because the maps (and conditions) are different.

round based on the robot's predictability, dependability, faith, and trust. Then, based on the measure of trust, a new robot's task is shown to them. In the user study to localize user trust, we will associate each level with a specific discretization of this range. Specifically, we will consider the following intervals for each of the levels, $\{[0, 0.25], (0.25, 0.5], (0.5, 0.75], (0.75, 1]\}$. Depending on the condition the participant belonged to, they are either shown an action selected by a policy calculated from our method (for Trust-aware condition), or an explicable plan (for Always explicable condition) or is randomly shown either the optimal or explicable plan with an equal probability (for Random Policy condition).

**Human Subjects** We recruited a total of 62 participants, of whom 38% were undergraduate, and 62% were graduate students in Computer Science, Engineering, and Industrial Engineering at our university. We paid them a base of $10 for the study and a bonus of 1¢ per point, given the total points they will get in ten rounds. From the participants, 24 were assigned to the trust-aware condition, 18 to always explicable condition, and finally 20 to the random policy condition. Then, we filtered out any participants who monitored the robot in less than four rounds because they wouldn't have monitored the robot long enough to have a correct expectation in regards to the robot behavior.

**Results** Across all the three conditions, we collected (a) participants' trust measures in each round, (b) robot's total plan execution cost, and (c) participants' monitoring cost. For the monitoring cost, we consider the minutes participants spent on monitoring the robot in each round, which was approximately 3 minutes for each round of monitoring. As shown in Figure 2, we can see that the total cost (the robot's plan execution cost and the participant's monitoring cost) when the robot executes trust-aware behavior is significantly lower than the other two cases which means that following trust-aware policy allows the robot to successfully optimize the team performance. From Figure 3, we also observe that the trust (as measured by the Muir questionnaire) improves much more rapidly when the robot executes trust-aware policy as compared to the random policy. Though the rate for the trust-aware policy is less than the always explicable case, we believe this is an acceptable trade-off since following the trust-aware policy does result in higher performance. Also, we expect trust levels for trust-aware policies to catch up with the always-explicable conditions over longer time horizons.

**Statistical Significance–**We tested the two hypotheses by performing a one-tailed p-value test via t-test for independent means with results being significant at $p < 0.05$ and find that results are significant for both hypotheses. 1) For the first hypothesis H1, we tested the mean cost with the null hypothesis of team performance cost has a mean of 3170.199 (using data from the "always explicable" scenario), we find that p-value is less than $0.00001$. 2) For the second hypothesis H2, we tested the mean trust value for the last round and mean value over last two rounds with the null hypothesis being the trust value has a mean of $0.5458$ and

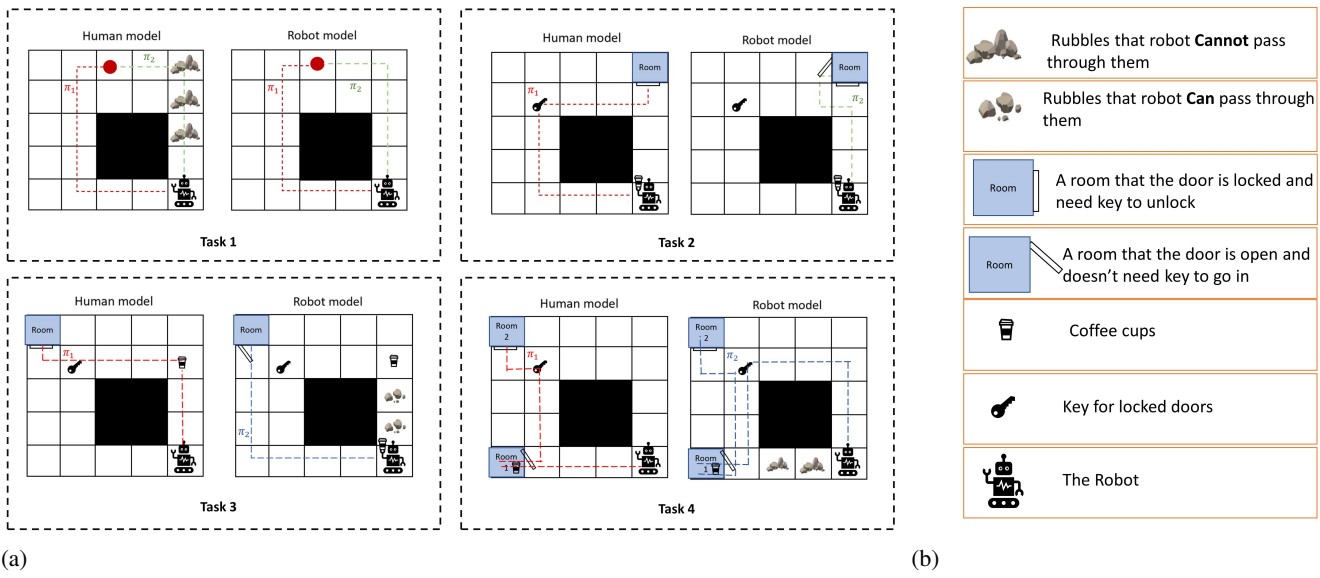

(a)                                                                                    (b)

Figure 1: (a) The human and the robot model of the map for the four different tasks, (b) The map description

0.5416 in the last round and and last two rounds respectively (using the data from "Random" scenario), we find that the p-values are $0.03174$ and $0.03847$ respectively. So, the results are statistically significant and show the validity of our hypotheses.

Also, we ran Mixed ANOVA test to determine validity of second hypothesis H2, and we found that there was a significant time (round)[5] by condition interaction $F(1,27) = 4.72$, $p = 0.039$, $\eta_p^2 = 0.15$. Planned comparison with paired sample t-test revealed that in participant in Trust-Aware condition, trust increases significantly in round 10 compare to round 1, $t = 3.55$, $p = 0.002$, $d = 0.84$. There was however no difference in trust increase between round 1 and round 10 in the Random Policy condition $t = -0.15$, $p = 0.883$, $d = -0.046$. Both of these results follow our expectation about the method. Moreover, we ran Mixed ANOVA test on Trust-Aware vs. Always Explicable condition to check trust evolution over time, and we found that there was no significant time (round) by condition interaction $F(1,26) = 2.21$, $p = 0.149$, $\eta_p^2 = 0.08$. Planned comparison with paired sample t-test revealed that in participant in Trust-Aware condition, trust increases significantly in round 10 compare to round 1, $t = 3.55$, $p = 0.002$, $d = 0.84$. There was also significant difference in trust increase between round 1 and round 10 in the Always Explicable condition $t = 5.04$, $p = 0.001$, $d = 1.59$. This seems to imply that there isn't a significant difference between our Trust-aware method (which is a lot more cost efficient) and Always Explicable case with regards to engendering trust.

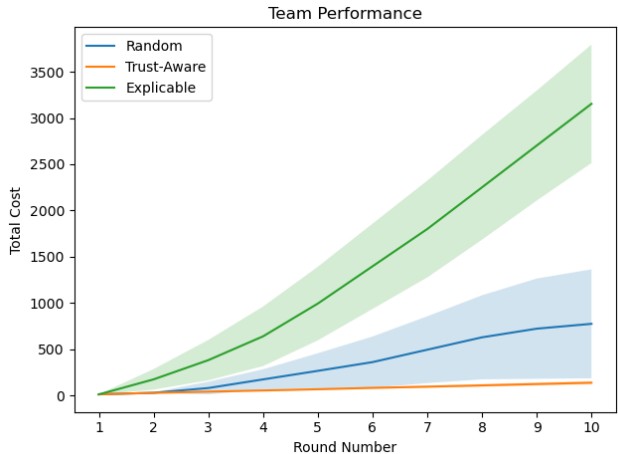

Figure 2: Team performance as cumulative plan execution cost and participants' monitoring cost (Mean $\pm$ std of all participants).

## Conclusion and Discussion

In this paper, we presented a computational model that the robot can use to capture the evolution of human trust in longitudinal human-robot interactions. This framework allows the robot to incorporate human trust into its planning process, thereby allowing it to be a more effective teammate. Thus our framework would allow an agent to model, foster, and maintain the trust of their fellow teammates. Thereby causing the agent to engage in trust engendering behavior earlier in the teaming lifecycle and be able to leverage trust built over these earlier interactions to perform more efficient but potentially inexplicable behavior later on. As our experimental studies show, such an approach could result in a

---

[5]We considered the change over first and last rounds

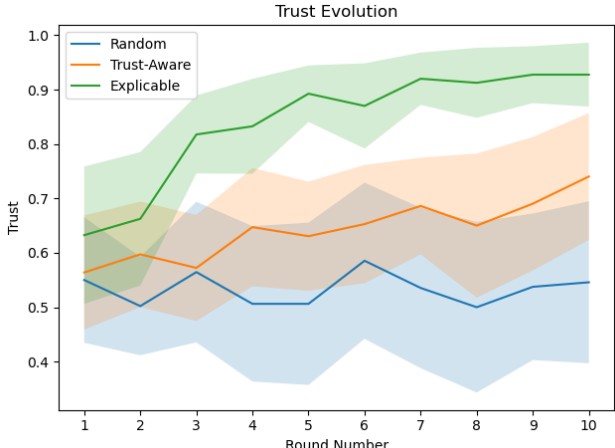

Figure 3: Trust evolution (as measured by the Muir questionnaire) through robot interactions with participants (Mean ± std of all participants).

much more efficient system than one that always engages in explicable behavior. We see this framework as the first step in building such a longitudinal trust reasoning framework. Thus a natural next step would be to consider POMDP versions of the framework, where the human's trust level is a hidden variable. We also plan to investigate methods to effectively learn the various parameter of our Meta-MDP or perform direct RL over this MDP.

**Acknowledgments.** This research is supported in part by ONR grants N00014-16-1-2892, N00014-18-1-2442, N00014-18-1-2840, N00014-9-1-2119, AFOSR grant FA9550-18-1-0067, DARPA SAIL-ON grant W911NF-19-2-0006, NASA grant NNX17AD06G, and a JP Morgan AI Faculty Research grant.

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
