# OpenReview forum: "Trust-Aware Planning:Modeling Trust Evolution in Longitudinal Human-Robot Interaction"
_icaps-conference.org/ICAPS/2021/Workshop/XAIP — XAIP 2021_

### Official Review · AnonReviewer1 · 2021-06-26
**It is clear that a lot of work has gone into this research and the writing of this paper. This would benefit from the discussion within the workshop to further improve it. I would recommend this paper be accepted.**

**Rating:** 8
**Confidence:** 4

**Review:**

**Summary:** This paper presents a novel computational model to integrate trust in longitudinal human-robot interaction. By capturing the trust within the model, robots can decide whether to build the trust by providing more explicable plans/explanations or focus on maximizing the team goals. The authors validate their model with a human-subject study.

**Strength:**

(+) The paper is well written and provides a solid motivation theory behind their work.

(+) The authors provide clear mathematical background on human-aware planning, making this work easily accessible to newer researchers. I also really liked how they have separate sections on assumptions that are well laid out.


**Minor comments:**

- Not clear from the experimental design whether it is within or between subjects or something else?

- Task one in figure 1 has different representations compared to other tasks (i.e., explicable and optimal plan)

**Suggestions and Questions to the Authors:**

1. Work claims the modeling trust in the longitudinal human-robot interaction, but the interaction is pretty short (3 minutes for each round). I am concerned about how this work would test against extended human-robot interactions spaced over weeks/months. Generally speaking, longitudinal human-robot interaction refers to studying the same people over a long period of time. I would be careful while phrasing it as longitudinal HRI

2. I am interested in knowing the author's opinion on how trust builds throughout interaction (related to the above-mentioned first point). Precisely, how much leverage do robots have to build trust over a single short interaction vs. single long interaction vs. spaced repetitive longitudinal interaction?

3. One of the challenges with this work is the accessibility of trust over a period of time (Assumption R3). The authors suggest that we could either get those by directly asking the supervisor or querying the users. Some of the foreseeable challenges are: 1) converting those subjective trust levels into quantifying values that the agent can use, and 2) We can't expect supervisors/collaborators to be continuously available or open to query which has cost associated with it and can lead to irritation/mistrust regularly. I would like to know the author's thoughts on these challenges or be clarified within the limitations of this work.

4. I think there is some similarity between this work and work done by Nikolaidis et al. in the adaptive model human-robot interaction. I would be interested to see a comparison or authors thought on the same in related work

- Nikolaidis, Stefanos, David Hsu, and Siddhartha Srinivasa. "Human-robot mutual adaptation in collaborative tasks: Models and experiments." The International Journal of Robotics Research 36.5-7 (2017): 618-634.
- Nikolaidis, Stefanos, et al. "Game-theoretic modeling of human adaptation in human-robot collaboration." Proceedings of the 2017 ACM/IEEE international conference on human-robot interaction. 2017.
- Nikolaidis, Stefanos, et al. "Human-robot mutual adaptation in shared autonomy." 2017 12th ACM/IEEE International Conference on Human-Robot Interaction (HRI. IEEE, 2017.

5. Authors use “P” to represent the state transition function instead of “T” which is what I have seen generally papers. Is there a reason for switching?

---

### Official Review · AnonReviewer2 · 2021-06-29
**Interesting and promising idea, lacks grouding/validation of the trust model**

**Rating:** 7
**Confidence:** 4

**Review:**

I enjoyed the reading the paper and believe it is interesting and important work that will definitely inspire further research.
The paper is clear and mostly well organized.

My main concern is the lack of grounding and validation of the trust model.
The approach heavily relies on having access to a good model of trust, and therefore I was disappointed to see the way the model is derived. The model does not seem to be grounded on previous work on the psychology of trust, nor on data from a preliminary user study. It has a large amount assumptions (transition function equations, discretization of trust, the fact that trust goes up or down one level at a time and never directly to zero, etc.) The model should also have been validated directly from the user study data, which I believe should be relatively straightforward given the data that the user study already provides (trust measurements and what the users saw at each point) --- i.e. how well does the model match what we see in Figure 3?

Another important limitation is the lack of use of explanations in the user study --- thus raising the question of why this functionality was introduced in the first place (in this paper).

A final concern is with the user study setup, that is currently not clear in some respects --- see questions below.

However, I think the paper is still relevant and worthy of being at the workshop as long as some discussion and detail is added.

Suggestions:

1) If there was some work that was used as inspiration to arrive at this trust model (e.g. psychology of trust? previous robot trust papers?), then please explicitly state so. Furthermore, if possible please validate the trust model from the trust data already gathered in the user study (e.g. error between users' trust trajectory what is assumed by the authors' transition function; distance between w(i) used in the method and what was observed in the data). If this is not possible, then please discuss why not, or discuss the need for doing this in future work.

2) The paper is well organized, but there is one small exception which is the "remark" sub-section that discusses issues of correctness of the model and automation bias - I believe this discussion belongs in the conclusion/discussion section and perhaps also in the introduction. Please re-organize this bit.

3) [for future work] Start by conducting a user study to gather trust data (or use the data already gathered if it is enough), and build a model of trust from the bottom-up; and then re-run the user study with this new model, and validate the model again on the data from the new study. ---> this would probably have been a more sound way of doing this paper.

4) [for future work] Include the use of explanations in the user study.

5) [for future work] Include a "always optimal" baseline

Minor suggestions:
- Make the paper less casual at points like "[we] changed it a little bit" (I would just state the changes) and "to make sure it does its job" (I would say what is meant by this exactly, i.e. to monitor the robot and interrupt it if they think that is necessary).
- "the map is shown to the participants are" ---> "the map that is shown to the participants is"

Questions for the authors:

a) Are participants actually shown the rocks/keys shown in the figure even when they don't exist? If so, how do you justify to the users that they were given a map that was wrong? (for example when the robot goes over what were supposed to be rocks). I believe this will have a crucial influence on user behavior and trust. It also needs to be clarified and discussed in the paper.

b) If users choose to monitor, do they still see their own (wrong) maps or are the maps updated once the robot crosses a rock that was not there, for example? It feels like a very unnatural setting to just let a robot go over an obstacle, or go through a door that was supposed to be closed, and to have a static map over a full experiment.

c) When users decide to interrupt a robot, are they told whether that interruption was needed somehow?

d) If the robot's map is always correct, then the optimal strategy is to never monitor robot, right? If so then it is important to see longer term results and understand whether users ever choose to monitor after a long time (i.e. is there automation bias?), and then to consider what to do to avoid that.

e) Why is your p-value the same for both the last round and last three rounds? I would say this would be virtually impossible. Please discuss (or verify whether there was no mistake involved).

---

### Meta-Review · Area_Chairs · 2021-07-07

**Recommendation:** Accept
**Confidence:** 5

**Metareview:**


Thanks for your submission to the workshop!

Summary: This paper presents a framework for adapting the level of explicability/explanation of an agent's plans (with respect to a supervisor), as the supervisor's trust in the agent varies. The reviewers agree that the paper presents interesting work, and that the topic is relevant to the workshop.

Strengths:
* The paper is well written, clearly presented and well motivated;
* The topic is important and very relevant to the topic of the workshop;
* The concepts used in the work are clearly presented;
* A user study was conducted and provides support for the work.

Limitations:
* A key concept within this work is the trust model of the supervisor. The current presentation provides little support for the chosen model of trust. It is unclear how existing models of trust could be captured in the presented framework;
* Similarly, the paper does not address the choice of granularity for the trust model (i.e. problem not action);
* The current pathway to a usable system is unclear (requirement of measure of user trust)
* Casual writing, including contractions.

Overall, novel work that can be a interesting contribution to the workshop. We hope that you find the reviewers' comments useful and that they will help you to revise your paper.

---

### Decision · Program_Chairs · 2021-07-08

Accept